# Optically-biased Rydberg microwave receiver enabled by hybrid nonlinear interferometry

Sebastian Borówka [1,2], Mateusz Mazelanik [1], Wojciech Wasilewski[1,2] & Michał Parniak [1,2] ✉

Coupling a Rydberg vapour medium to both microwave and optical fields enables the benefits of all-optical detection, such as minimal disturbance of the measured field and resilience to very strong signals, since no conventional antenna is required. However, peak sensitivity typically relies on adding a microwave local oscillator, which compromises the all-optical nature of the measurement. Here we introduce an alternative, *optical-bias detection*, that maintains fully optical operation while achieving high sensitivity. To address laser phase noise, which is critical in this approach, we perform a simultaneous measurement of the noise using a nonlinear process and correct it in real time via data processing. This yields a 35 dB improvement in signal-to-noise ratio compared with the basic method. We demonstrate a sensitivity of $176\,\mathrm{nV/cm/\sqrt{Hz}}$, reliable operation up to 3.5 mV/cm at 13.9 GHz, and quadrature-amplitude modulated data transmission, underlining the ability to detect microwave field quadratures while preserving the unique advantages of all-optical detection.

Rydberg atoms facilitate strong interactions, which enable current breakthroughs in quantum computing[1]. The large dipole moments associated with transitions between different Rydberg states, which govern the strength of these interactions, also make the atoms very sensitive to external fields. This metrological property, particularly valuable for sensing microwave (MW) fields, has long been recognised in atomic beam and cold-atom experiments[2]. However, only recent advancements have made practical quantum metrology feasible using hot-atom vapour cell systems.

The original approaches to hot-atom sensors were based on the splitting of an electromagnetically induced transparency (EIT) line due to the MW-induced Autler-Townes (A-T) effect. This elegant and simple approach[3–5] enables self-calibration based on atomic constants but only allows for a limited sensitivity and does not yield phase information. Since then, progress has been made, extending this method to facilitate imaging[6,7], provide phase information and enhance sensing[8–13] or enable applications in communication protocols[14–22], as well as miniaturisation of the receivers[23,24].

The sensitivity metric has been further enhanced by treating the atoms as a MW mixer with optical output[25,26], effectively constituting a superheterodyne receiver (superhet)[27]. Subsequent developments allowed consecutive enhancements of the method[28–34] and adapting it to specific problems, such as measurements of angle of arrival[35] and polarisation[36].

Atomic superhet reception provides an important link to standard radio communication, where the superheterodyne detection method is present in all modern receivers. In the end, however, while the readout is optical, the Rydberg-atom receiver still requires a strong MW local oscillator (LO) for operation. In terms of practical applications, this solution is not ideal, as a MW antenna needs to be part of the receiver, rendering some of the advantages of atomic sensors, such as weakly disruptive (stealthy) measurement of the electric field[37], irrelevant.

[1]Centre for Quantum Optical Technologies, Centre of New Technologies, University of Warsaw, Warsaw, Poland. [2]Faculty of Physics, University of Warsaw, Warsaw, Poland. ✉e-mail: mparniak@fuw.edu.pl

Drawing inspiration from our recent discoveries in MW-to-optical conversion in Rydberg atoms[38], we extend the well-established two-photon Rydberg excitation scheme by the addition of two optical fields in the near-infrared, realising a three-photon Rydberg excitation scheme via the practically Doppler-free $5D_{5/2}$ energy level in rubidium, already explored in some earlier works[39,40]. In this approach, the additional optical fields play the same role as the MW LO in superhet detection: they induce a beat-type modulation in the probe field transmission, and the measurement of its amplitude serves as a MW electrometer. Similar approaches have been presented with the use of two MW fields[41,42], where several all-optical solutions were also mentioned but not realised. The most important challenge arising in this all-optical approach is the collective laser phase noise, which transfers into the modulation signal. In the superhet detection, this issue is absent, as the MW LO enables phase stabilisation for the process, independently from the stability of the optical fields. However, it is no longer possible in an all-optical scheme.

In this work, we demonstrate an all-optical Rydberg-atom receiver, based on biasing the system only via optical fields ("optical-bias" detection), that eliminates the need for a MW LO in the detector. Instead, in this realisation the MW reference LO is detached from the cell sensor and is only part of the electronic subsystem of the setup. Furthermore, we overcome the challenge of laser phase noise by measuring it through a difference frequency generation (DFG) process using the same laser fields, and then removing it via real-time correlation on a field-programmable gate array (FPGA). This renders signal free of laser noise and achieves performance directly comparable to superhet detection.

## Results

### Comparison between superhet and optical-bias MW detection setups

**Conventional Rydberg superhet.** Let us start by recalling the standard atomic superhet detection scheme. In the superhet detection, pictured schematically in Fig. 1a, the optical fields, probe ($E_p$) and coupling ($E_C$) beams counter-propagate through an atomic cell, enabling precise detection of a MW signal field ($\mathcal{E}$). Like any fast power detector this setup can be used as a mixer to detect a weak $\Delta$-detuned MW signal at frequency $\omega_S = \omega_0 + \Delta$ using a strong resonant MW LO ($E_{LO}$) at frequency $\omega_{LO} = \omega_0$, where $\omega_0$ is the frequency of atomic transition. When both fields are present, the total MW intensity beats at the difference frequency $\Delta = \omega_S - \omega_{LO}$ in the radio frequency (RF) range. This signal is, in turn, transduced to the optical probe field via the Rydberg EIT effect[25]:

$$\delta T_p \sim |E_C|^2 \mathrm{Re}(E_{LO}^*\mathcal{E}) \propto \cos(\Delta t - \phi_{LO} + \phi_S). \quad (1)$$

With monochromatic fields, the signal is directly at frequency $\Delta$, and depends on the relative phase between LO ($\phi_{LO}$) and signal ($\phi_S$) – see Fig. 1b. Remarkably, the phase of probe and coupling lasers is cancelled out in the expression, as the signal only depends on their intensities. For now, we will also neglect any phase delay introduced by the atomic medium. In general, the superheterodyne (photocurrent) signal is proportional to $\mathcal{I}_S \propto |E_p|^2|E_C|^2\mathrm{Re}(E_{LO}^*\mathcal{E})$. We demodulate the signal using analogue-digital conversion (ADC) followed by a digital IQ mixer (which is a typical approach in modern software-defined radios – SDR) in an FPGA architecture, obtaining

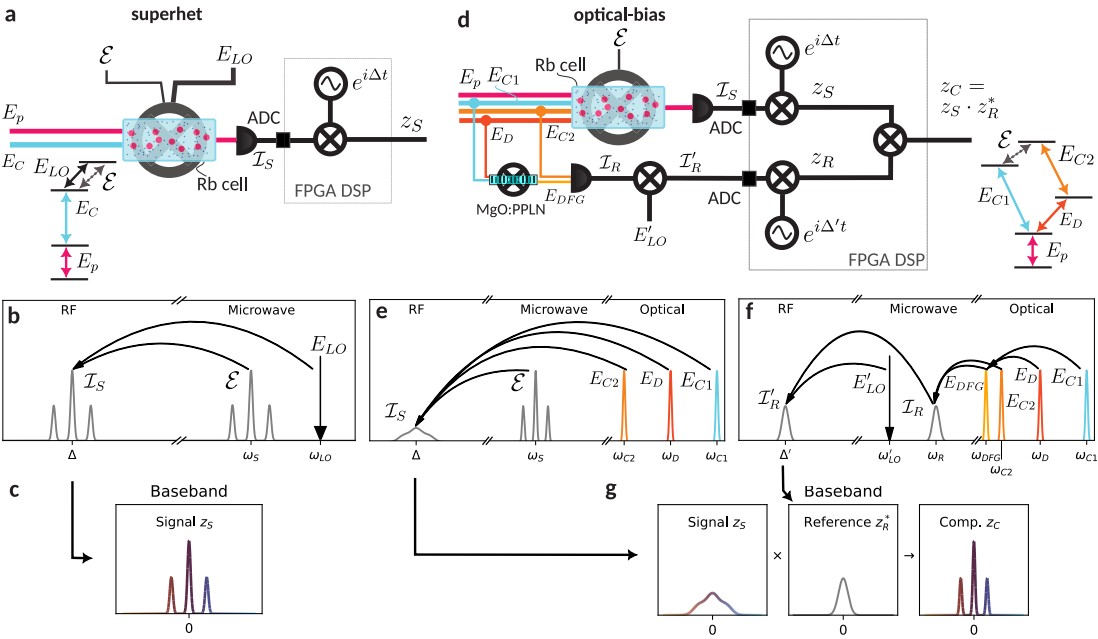

**Fig. 1 | Comparison between superhet and optical-bias ideas. a** Operating principle of a MW superhet receiver. The atoms act as a microwave mixer, with optical fields, $E_p$ and $E_C$, having a role only in the detection. The signal $\mathcal{E}$ is thus mixed with the LO field $E_{LO}$, transduced to the RF domain $\mathcal{I}_S$, detected with a photodiode, converted to digital data with analogue-digital converter (ADC), and then demodulated to a complex baseband signal $z_S$ via FPGA digital signal processing (DSP). **b** In the spectral picture the atoms act as a mixer between the MW signal $\mathcal{E}$ (represented as a modulated three peaks feature) and the LO $E_{LO}$, yielding a signal in the RF range $\mathcal{I}_S$, centred at the detuning $\Delta = \omega_S - \omega_{LO}$. **c** At the end, a digital IQ mixer recovers the original modulation as a complex signal $z_S$. **d** Operating principle of a MW optical-bias receiver. The optical fields, $E_{C1}, E_{C2}$ and $E_D$, now have a primary role in the mixing process. An additional measurement of the combined

optical phase noise $\mathcal{I}_R$ (realised via difference frequency generation (DFG) of the optical fields in MgO:PPLN nonlinear crystal) allows the phase compensation. **e** The atoms now act as a mixer between the MW signal and optical fields. Because of the optical phase noise, the resulting photodiode (PD) signal in the RF range $\mathcal{I}_S$, centred at $\Delta = \omega_{C1} + \omega_S - \omega_D - \omega_{C2}$, is noisy and degraded. **f** To solve the problem with the noise, a reference path mixes the optical fields with a MW LO′, using a DFG process, a fast PD, and a mixer, in a three-step process. The reference signal $\mathcal{I}_R'$ contains all required information about the optical phase noise. **g** At the end, both the signal $z_S$ and the reference $z_R$ are IQ-mixed to a baseband complex signal, and correction is applied as a digitally-implemented complex multiplication. This results in the original modulation being fully recovered in the $z_C$ complex signal.

the full complex waveform:

$$z_S \propto E_{LO}^* \mathcal{E} e^{-i\Delta t}, \tag{2}$$

thus allowing the recovery of any original modulation imposed on the signal. The final signal is at baseband (see Fig. 1c).

In the general case, any signal represented by its two-sided power spectral density $S_\mathcal{E}(\omega)$ is thus first brought down to the RF domain $S_{\mathcal{I}_S}(\omega) = S_\mathcal{E}(\omega - \omega_{LO}) + S_\mathcal{E}(-\omega + \omega_{LO})$, under the assumption of a noiseless and single-tone LO. We demodulate this signal using an IQ mixer, which yields a final signal that directly represents the original complex field:

$$S_{z_S} = S_\mathcal{E}(\omega - \omega_{LO} - \Delta), \tag{3}$$

which shall be our ideal reference situation for further comparisons.

The role of the LO field in this kind of detection is dual: not only does it provide phase-reference, but also biases the atomic medium to the detection point, where the sensitivity is the greatest[27]. Typically, we select the central frequency of the signal $\omega_S$ such that $\Delta$ is of the order of a few MHz (RF range), thus avoiding technical low-frequency noise and assuring that $\Delta$ is larger than the signal bandwidth.

**Optical-bias (all-optical superhet).** Let us now bring the attention to the optical-bias detection, which we introduce here, pictured schematically in Fig. 1d. Here, instead of the MW LO field, we introduce two additional optical fields. Overall, the fields together with the MW signal form a loop of transitions. The probe signal normally experiences a combined effect of all optical fields, which resembles EIT. With the presence of the MW signal, the loop is closed, and the microwave signal is likewise transduced to the probe field:

$$\delta T_p \sim \mathrm{Re}(E_{C1}E_{C2}^*E_D^*\mathcal{E}) \propto \cos(\Delta t + \phi_{C1} - \phi_{C2} - \phi_D + \phi_S). \tag{4}$$

Thus, the detection stage is conveniently decoupled from the volume containing the MW field. Again, the beating transfers to the modulation of the probe field at the frequency

$$\Delta = \omega_{C1} + \omega_S - \omega_D - \omega_{C2}, \tag{5}$$

which here is understood as the mismatch in energies between the interacting fields, or a "fracture" of optical-atomic loop[43]. The final signal in the RF domain (see the Fig. 1e) at the photodiode (PD) takes on the form $\mathcal{I}_S \propto |E_p|^2 \mathrm{Re}(E_{C1}E_{C2}^*E_D^*\mathcal{E})$. The product of the three fields $E_{C1}E_{C2}^*E_D$ takes on the role of the LO.

Here, we are no longer able to assume that the fields are noiseless, as the signal includes a combined phase of three laser fields $\phi_{C1} - \phi_{C2} - \phi_D$. Overall, the final spectrum has the form $S_{\mathcal{I}_S}(\omega) = S_{E_{C1}} * S_{E_{C2}} * S_{E_D} * S_\mathcal{E}$, where $*$ denotes a spectral-domain convolution. The final spectrum is thus significantly broader, as its width is approximately the root sum squared of the linewidths of all three lasers.

A brute-force solution to this issue would be to use very narrow-linewidth lasers, which come with significant effort, expenses, and complexity. In this work, we use a separate nonlinear process, in which fields $E_{C1}$ and $E_D$ are mixed to obtain an optical field $E_{DFG} = E_{C1}E_D^*$—see the Fig. 1f. This field beats at a reference detector with the $E_{C2}$ field, yielding a microwave-domain photocurrent $\mathcal{I}_R \propto \mathrm{Re}(E_{DFG}E_{C2}^*)$. Then, the reference signal is mixed with the actual MW LO' ($E_{LO'}$), which is only present here and not in the core of the MW detector cell. This leads to an overall RF-domain signal:

$$\mathcal{I}_R' \propto \mathrm{Re}(E_{LO'}E_{C1}E_D^*E_{C2}^*) \propto \cos(\Delta' t + \phi_{LO'} + \phi_{C1} - \phi_D - \phi_{C2} + \phi_i), \tag{6}$$

where $\Delta' = \omega_D + \omega_{C2} - \omega_{C1} - \omega_{LO'}$ is selected close to $\Delta$ but not the same. Remarkably, this reference signal includes the same optical phase $\phi_{C1} - \phi_{C2} - \phi_D$ as before. The additional phase term $\phi_i$ arises from differences in optical paths of the fields contributing to the $\mathcal{I}_S$ and $\mathcal{I}_R'$ signals, i.e. it represents the phase of the hybrid interferometer.

The $\mathcal{I}_S$ and $\mathcal{I}_R'$ signals are now both in the RF domain. Offsetting of the signals from the zero frequency is necessary to recover their complex dependencies, $z_S$ and $z_R$, via the subsequent IQ mixers. In particular, this allows us to trace the laser phase as the argument of the full complex number $z_R$. In other words, we can distinguish laser frequency drifts to both negative and positive frequencies. At the same time, offsetting of the $\mathcal{I}_S$ signal by $\Delta$ is helpful, as it avoids low-frequency technical noises, similarly as in the standard superheterodyne detection. The offsets are selected to be different, i.e. $\Delta \neq \Delta'$, to avoid cross-talks between parts of the MW setup.

After the analogue-digital conversion (ADC), both of the $\mathcal{I}_S$ and $\mathcal{I}_R'$ are demodulated digitally at their central frequencies to basebands—see Fig. 1g. Demodulation results in complex signals $z_S$ and $z_R$. The final compensated complex signal is effectively generated as a product of the main signal and the complex conjugate of the reference signal:

$$z_C = z_S z_R^* \propto |E_p|^2 |E_{C1}|^2 |E_{C2}|^2 |E_D|^2 E_{LO'}^* \mathcal{E} e^{i(\Delta - \Delta')t}. \tag{7}$$

Remarkably, thanks to the complex conjugation, the phase noise is removed and the spectrum again resembles the ideal situation of Eq. (3). In other words, the optical phase $\phi_{C1} - \phi_{C2} - \phi_D$ between the two signals is cancelled out, we are only left with the interferometer phase $\phi_i$, which is much more stable by itself.

## Implementation

The rubidium energy levels employed in the presented optical-bias detection are shown in Fig. 2a. The standard two-photon Rydberg excitation at 780 nm (probe field $E_p$) and 480 nm (coupling field $E_C$ in superhet, and $E_{C1}$ in the all-optical scheme) is supplemented with two fields in near-infrared, 776 nm ($E_D$) and 1258 nm ($E_{C2}$), both of which are convenient to work with in terms of fibre-based solutions. All of the fields are resonant, apart from the $\Delta_{5D}$ detuning from the $5D_{5/2}$ and the beat note detuning set to $\Delta = 1.8$ MHz. We experimentally found the optimal working point when $\Delta_{5D} = -\Delta$ for small detunings, as indicated in Fig. 2a (for the full consideration of the choice of frequencies, consult Supplementary Section S.1).

To register the reference signal $z_R$, we take advantage of an optical nonlinear process. As shown in Fig. 2b, we realise a DFG between 480 nm and 776 nm, which yields a field at 1258 nm ($E_{DFG}$). The DFG field is combined with the primary 1258 nm field $E_{C2}$ on a fast photodiode, yielding a beat signal $\mathcal{I}_R$ around 13.9 GHz. This beat reference signal is then downconverted to $\Delta' = 4.6$ MHz by combining with $E_{LO'}$ on a microwave mixer.

## Spectral characteristics

For the first demonstration, we choose a monochromatic, narrowband microwave signal $\mathcal{E}$ with an amplitude of 720 $\mu$V/cm. The power spectrum of the uncompensated signal $\mathcal{I}_S$ in the optical-bias method is presented in Fig. 3a. The data is normalised to the carrier frequency peak centred around frequency $\Delta$, and the noise floor is dominated by the shot noise of the probe optical field (for the full consideration consult Supplementary Section S.2). The spectral FWHM (full width at half maximum) of the signal is 62 kHz, owing to the collective optical noise of the three contributing lasers. The power spectrum exhibits a characteristic structure with a main peak and a pedestal, which is due to laser stabilisation loops. Similarly, in Fig. 3b we show the power spectral density of the reference signal $\mathcal{I}_R'$. The signal, centred around $\Delta'$, exhibits the same features of the laser noise but with a significantly higher signal-to-noise ratio.

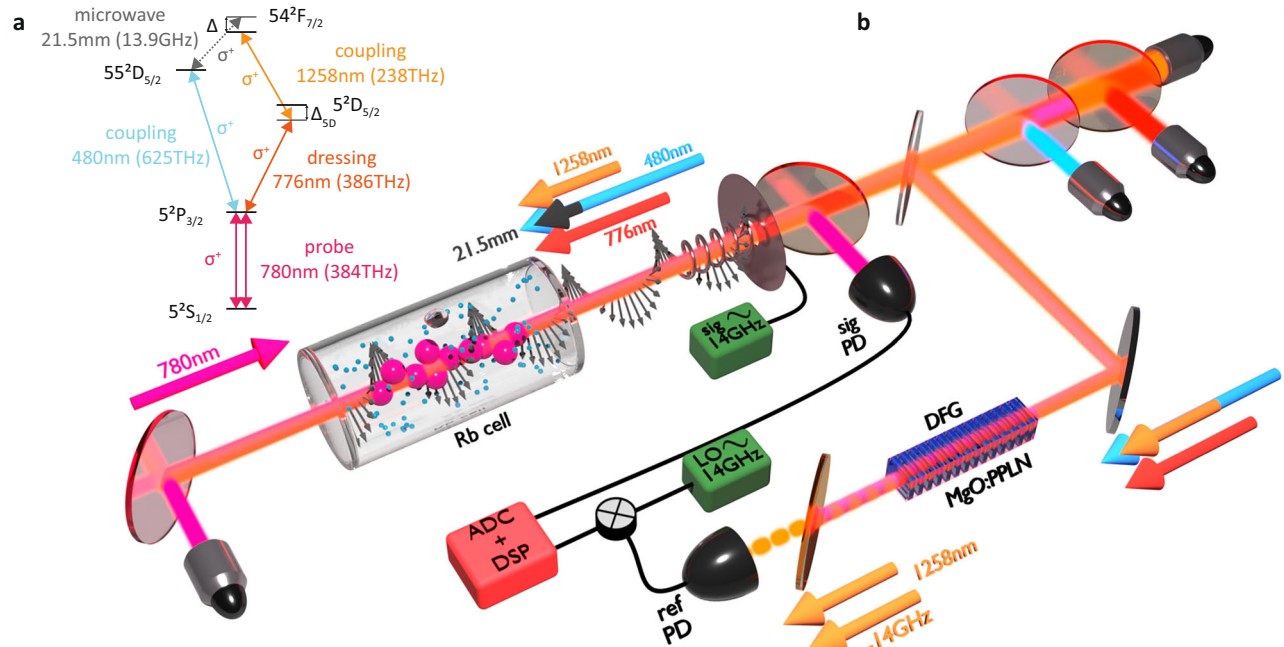

**Fig. 2 | Optically-biased MW receiver. a** Energy level structure utilised in the optical-bias detection. Two-photon (*probe–coupling*) and three-photon (*probe–dressing–coupling*) Rydberg excitation paths are used to access both energy levels connected by the MW transition. The $\sigma^+$ transitions ensure the largest transition dipole moments. All of the optical fields are atomic resonant, apart from the indicated detuning $\Delta_{5D} = -1.8$ MHz. The beat modulation in probe transmission, i.e., the transduced signal, is also observed at $\Delta = 1.8$ MHz due to the detuning of the MW field. **b** Experimental setup of the optical-bias receiver. Three optical (480 nm, 776 nm and 1258 nm) fields are divided into two paths. In the first path, they counter-propagate with respect to the 780 nm probe field, enabling partial Doppler effect cancellation. The laser fields are propagated as Gaussian beams focused to waists of around $w_0 = 250$ μm. They have matched circular polarisations and are combined inside $^{85}$Rb vapour cell using dichroic mirrors and spectral filters, while the signal is detected in the signal photodiode (PD). The second path leads to a difference-frequency generation (DFG) setup for laser phase spectrum detection. The 480 nm and 776 nm fields induce DFG at 1258 nm shifted by the frequency of the detected MW field, as dictated by the conservation of energy. Combined with the non-shifted 1258 nm field used in the detection setup, they generate a beat-note at 13.9 GHz on a reference PD. Then, downmixing with the LO′ signal enables the retrieval of laser-noise spectrum shifted to lower frequencies. For measurements, the cell is placed inside a MW absorbing shield with a helical MW antenna acting as a signal source. In the case of superhet detection measured for comparison, the 776 nm and 1258 nm are switched off, and MW LO is combined with the signal at the antenna.

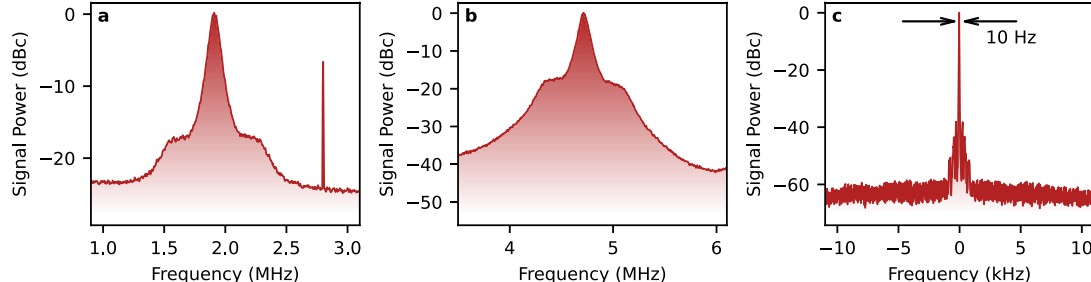

**Fig. 3 | Phase referencing the signal enables Fourier-limited spectral detection. a** Spectrum of the probe field modulation $S_{\mathcal{I}_S}(\omega)$ obtained in optical-bias detection of 720 μV/cm MW field. The maximum of the signal is 25 dB above the noise level, and the spectral width is 62 kHz FWHM. The $\mathcal{I}_S$ signal frequency is centred around $\Delta = 1.8$ MHz. The visible peak at ~ 2.8 MHz is a small superhet-type signal resulting from cross-talk. It is outside the set detection bandwidth. **b** Respective spectrum $S_{\mathcal{I}_R'}(\omega)$ of the reference signal obtained via DFG and subsequent beating with the 1258 nm laser. The $\mathcal{I}_R'$ signal frequency is centred around $\Delta' = 4.6$ MHz. The maximum of the signal is 41 dB above the noise level. **c** Respective spectrum of the phase-compensated signal $S_{z_c}(\omega)$. The maximum of the signal is 60 dB above the noise level. The spectral width of the signal is Fourier-limited at 10 Hz. The artefact spurs are at −38 dB below the signal. The spectra are estimated from experimental data using Welch's method.

The power spectrum of the phase-compensated signal $S_{z_c}(\omega)$ is presented in Fig. 3c. As the signal is complex, we present its two-sided spectrum around the zero frequency. The measurement duration chosen in this case and in the onward analysis is $t = 100$ ms, which corresponds to the resolution of 10 Hz. Thus, the noise-compensated signal is Fourier-limited to this resolution. The improvement in the signal-to-noise ratio, from 25 dB to 60 dB, yielding 35 dB, is consistent with the spectrum-based estimation limit, which is (62 kHz)/(10 Hz) = 38 dB.

The artefact spurs at the level of −38 dB below the signal are the results of imperfect balancing of MW detection and noise detection setups and can be eliminated with better alignment of the hybrid interferometer. To explicitly study the stability of this signal, we also perform a long measurement and estimate the Allan deviation, obtaining $\sigma(\tau = 1.4 \text{ s}) = 2.1$ Hz (consult Supplementary Section S.3 for a full plot of Allan deviation versus averaging time). To further improve resolution—if needed in a given application—we anticipate that better electronic solutions need to be employed, and the phase of the hybrid

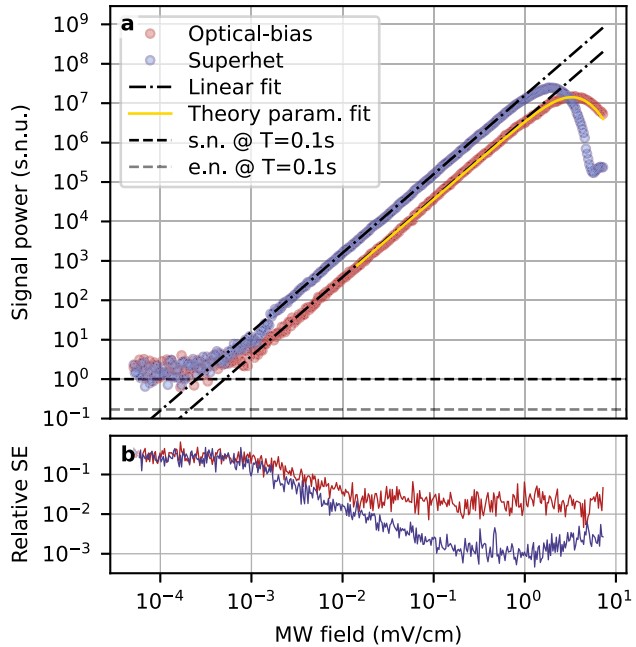

**Fig. 4 | Dynamic range of optical-bias parallels the superhet method.**
**a** Comparison between results obtained in superhet measurement (blue dots) and optical-bias (red dots). Both results are presented in relation to their respective detection noise levels (black dashed line), which in both cases is mainly the shot noise of the detected probe field transmission. This is denoted by the use of shot noise units (s.n.u.). The optical-bias measurement method results in comparable, though slightly worse, sensitivity and overall efficiency. Notably, however, it becomes saturated for larger MW fields than the superhet method, thus retaining a very similar dynamic range. The results presented here are averaged over $n = 8$ shots for each point to facilitate better comparison. The yellow solid line represents a theoretical prediction. Noteworthy, the saturation point is predicted accurately, and the experimental results differ from the theoretical predictions only in the stronger MW field regime, where the atomic response can no longer be considered instantaneous. The horizontal dashed lines represent the shot noise (s.n., black) and electronic noise (e.n., grey) levels. **b** Relative standard error (SE) of the data points from the Subfig (**a**).

interferometer $\phi_i$ has to be stabilised, e.g. with shortening interferometric arms' lengths of the system or using fibre solutions.

Note that in Fig. 3c we present only a part of the instantaneously acquired MW spectrum. In this realisation, the bandwidth is limited to 1 MHz due to the choice of demodulation frequencies. However, in full, we are able to observe the measurement in the 5.8 MHz FWHM MW bandwidth limited by the response of atoms (for the full consideration, consult Supplementary Section S.4).

## Dynamic range

By attenuating the MW signal, we measure the sensitivity of the optical-bias detection method. The results are presented in Fig. 4, where we facilitate comparison with the superhet method. To achieve adequate comparison, we normalise the signals to their respective shot noise of the probe optical field[44], which in both methods is the dominant noise component (for the full consideration of noise for optical-bias consult Supplementary Section S.2). The need for normalisation is only due to different gains employed in the DSP. In the optical-bias method, we find saturation due to energy level shifts at MW field 3.5 mV/cm, while the sensitivity reaches $176\,\mathrm{nV/cm/\sqrt{Hz}}$. These results are compared with the superhet method realised in the same setup, where we register the saturation at 1.84 mV/cm and the sensitivity of $87\,\mathrm{nV/cm/\sqrt{Hz}}$. Notably, the optical-bias method saturates for larger MW fields, while having

slightly worse sensitivity, though the dynamic range is almost equal at 76 dB for $t = 100$ ms measurement duration.

The behaviour of atomic saturation is predicted by the numerical simulation based on the theoretical model described in the "Methods" section. The results comparing the experimental data with the simulated function are presented in Fig. 4. In this case, the shape of the function is predicted with measured parameters, and the only free parameter is the constant scaling of the overall signal strength.

## Signal transduction in EIT spectra

To further elucidate the properties of the scheme and its differences from the standard atomic superhet, let us focus on the experimental analysis of the signal transduction in the probe field detuning $\delta_p$ domain, once again in the working point of 720 μV/cm MW signal field. In the Eqs. (1) and (3), we have not yet considered that the atomic response affects both the amplitude and phase of signal transduction $\mathcal{E} \to \delta T_p \to z_{(S/C)}$ in both cases. We will thus study a general complex transduced signal $z_{(S/C)}(\delta_p)$, given a constant $\mathcal{E}$.

We start with the superhet method, where in the probe field absorption, Fig. 5a, we observe a widened EIT feature (we found the optimal working point of MW LO to be at only $1.6 \cdot 2\pi$ MHz Rabi frequency for the transduced signal feature at $\Delta = 1.8$ MHz). Respectively, the transduced signal $z_S(\delta_p)$ is presented in Fig. 5d, bearing resemblance to the simple transfer of amplitude modulation of the MW field[13].

The optical-bias method exhibits a probe transmission spectrum resembling off-resonant A-T splitting due to MW field. However, this effect is only due to optical fields—see Fig. 5b. While the splitting in the spectrum is of the order of the coupling field Rabi frequencies, the structure is generally more complex than in the simple A-T splitting case. The visible EIT feature outside the ±10 MHz range is due to interaction with the sublevels not taking part in the optical-bias detection method, particularly the hyperfine splitting of the 5P$_{3/2}$ and 5D$_{5/2}$ levels. The amplitude of the transduced signal $|z_C(\delta_p)|$, Fig. 5e, is similar to the superhet method. However, the dependence of transduction phases $\arg(z_{(S/C)})$ on probe detuning differs in the direction of transitions.

The results obtained in the numerical simulation based on the theoretical model are shown in Fig. 5c, f. Both the probe field transmission and the amplitude of the signal are reproduced in terms of shapes in the probe field detuning domain. The difference in the direction of the transitions of the signal phase may be attributed to the interaction with other atomic sublevels visible in the experimental probe transmission, Fig. 5b.

## Quadrature-amplitude modulation

In previous sections, we sent a monochromatic MW signal to the atomic receiver. To demonstrate that our all-optical receiver can also receive modulated signals, we encode a pseudo-random 4-symbol Quadrature-amplitude modulation (QAM4) sequence in the MW field $\mathcal{E}$.

For this demonstration, we prepare a sequence of $8 \times 10^3$ QAM4 symbols and transmit them in the signal field at a rate of 122.07 kBaud. The symbols are received via the atomic ensemble and carry through the entire real-time processing chain. We can compare the power spectrum of the output signal with ($S_{z_C}$) and without ($S_{z_s}$) the phase compensation step, as shown in Fig. 6a. We observe that the compensation step is both necessary and sufficient to recover the modulation features. Next, we process the registered signal $z_C$ digitally and estimate the symbol at each time window from 32 raw data points per each symbol. We also employ a standard radio technique of carrier recovery (see Methods).

In Fig. 6b, we present an example estimate of the first ten symbols in the sequence. Next, all symbols are drawn in the IQ diagram in Fig. 6c and coloured according to the symbol sent. The presented

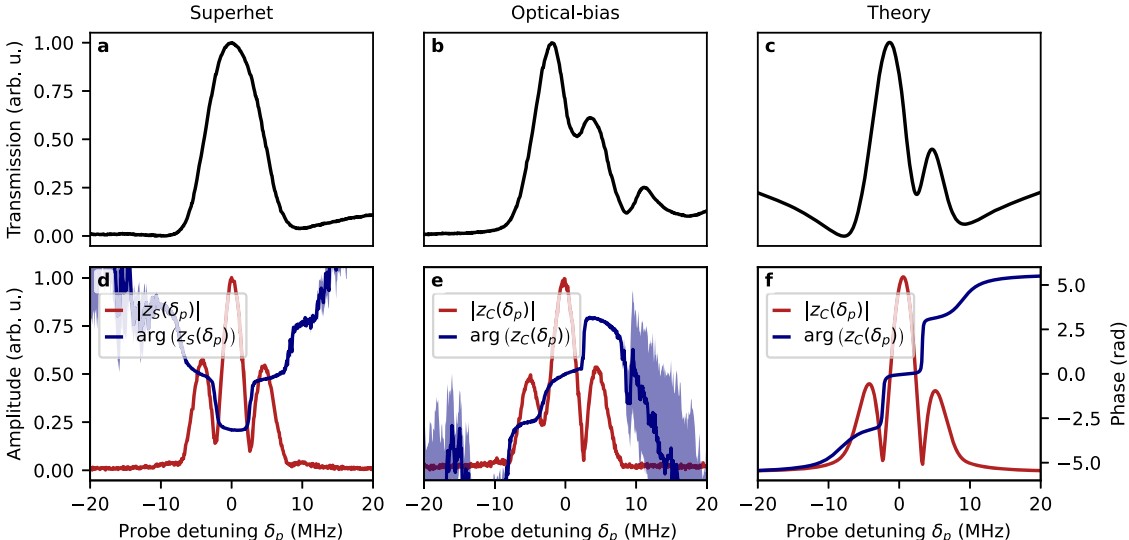

**Fig. 5 | Phase-sensitivity of the detection allows the study of signal transduction.** Comparison of EIT effects (upper row, **a**–**c**) and signal transduction (lower row, **d**–**f**) for superhet (left column, **a**, **d**) and optical-bias (middle column, **b**, **e**), and the theoretical prediction for optical-bias (right column, **c**, **f**) in the domain of probe field detuning $\delta_p$. For signal transduction both amplitude $|z_{(S/C)}(\delta_p)|$ (red lines) and phase $\arg(z_{(S/C)}(\delta_p))$ (blue lines) are shown. The MW field in all cases is 720 $\mu$V/cm. Notably, despite similarities in the shape of signal transduction amplitude, the optical-bias method exhibits a different direction of the transition of phase in the demodulated signal than the superhet method. The presented data is averaged over $n = 9$ separate measurements. The shaded blue regions represent the uncertainty (standard error) of the phase. The uncertainty of the measured transmission and amplitude in all cases is smaller than the thickness of the plot lines.

transmission achieves an errorless operation, strongly supporting the feasibility of both our approach and implementation for an all-optical receiver with a phase reference based on an auxiliary non-linear process.

## Discussion

We demonstrated a method of Rydberg atomic detection of MW fields that is both all-optical and sensitive, as shown by the comparison with the standard atomic superhet method. This paves the way to achieving all of the merits of all-optical measurement, i.e. weak disruption of the field and invulnerability to strong interaction, while not compromising on the sensitivity, and maintaining the phase-sensitivity of the superhet—a necessary condition for use in modern communication protocols. Furthermore, as the atomic medium interacts with a single MW frequency, it is possible to precisely design a resonant MW cavity around it, which would allow for surpassing the conventional MW receivers[45,46]. Notably, an alternative research path was undertaken to explore a Doppler-free scheme in the A-T splitting and also achieved considerable enhancements in sensitivity, while remaining all-optical[47]. We anticipate that the methods may converge to combine fully phase-sensitive reception with SI-calibrated measurements better.

We note that in our method, the optical fields together form an effective LO, which results in a modulation of the atomic coherence. This atomic polarisation may result in a weak stimulated emission of a MW at the LO frequency, but the power of this MW field is orders of magnitude lower than in the superheterodyne case. In essence, our method retains the all-optical operation by directly affecting the atomic coherence via optical fields.

In principle, the demonstrated optical-bias method can be extended to different Rydberg transitions, thus enabling the tunable detection of a wide bandwidth of MW and mmWave frequencies. We anticipate that the sensitivities and dynamic ranges achievable at different transitions will follow the relations already explored in the context of superhet measurements[11], which are largely determined by transition dipole moments and the choice of angular momentum of Rydberg states. However, the limitation of the presented realisation is the presence of the LO′ field $E_{LO'}$ at the microwave frequency needed

to downmix the phase reference signal, which presents a technological challenge at higher frequencies, especially above 100 GHz. This can be resolved by using an electro-optic modulator with high-order sideband modulation to shift one of the laser frequencies in the phase reference setup, thus avoiding the high-frequency LO′ field altogether and achieving another advantage over the standard superhet scheme. For example, to shifting the DFG field by $n$th-order modulation at frequency $\omega_{LO'}/n$ yields an immediate beat-note at an RF frequency $\Delta'$, without the need for the presence of the high LO′ frequency in the system. A similar frequency transfer could also be achieved with a relatively simple optical frequency comb (OFC) source. On the other hand, the optical-bias detection method can be massively improved with the use of phase-stable laser fields, e.g. with locking all of the lasers to an OFC. In this case, if the desired sensitivity and spectral resolution are achieved, the optical-bias method would work without the need for an additional reference signal $z_R$ obtained in the hybrid interferometer. Nevertheless, we predict that for most practical devices, the referencing protocol will be a significantly less resource-intensive solution.

Furthermore, here we have considered only the near-resonant detection of MW fields. While experimental investigation needs to be conducted in the future, we anticipate that the far-detuned detection schemes may follow the well-established techniques and tuning schemes for the superhet detection method[10,31,48], as both methods rely on wave-mixing processes. In the superhet method, it is enough to tune the microwave LO. In the method presented here, it would similarly be enough to tune the C2 coupling laser. In both cases, the signal can be detected, albeit with worse sensitivity. The caveat in the optical bias method is that for the best efficiency in the far-detuned detection, the $\Delta_{5D}$ detuning, and possibly other detunings, should be applied adequately (as exemplified in the bandwidth consideration in the Supplementary Section S.4).

Overall, we envisage that the all-optical receiver presented in this work, which would use a waveguide-based DFG or a fibre OFC, could constitute a compact and robust receiver with exceptional sensitivity, broad tuning, and all-optical detection, delivering on the most exciting promises of Rydberg-based quantum metrology.

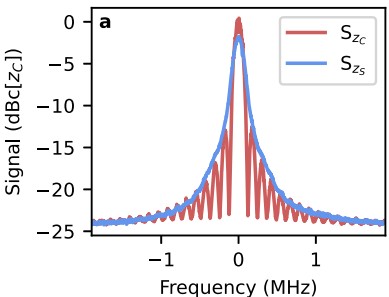
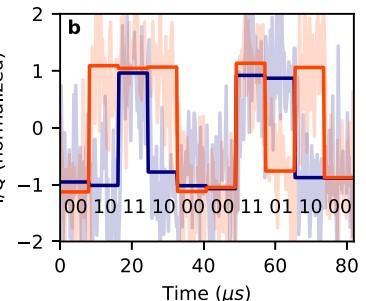
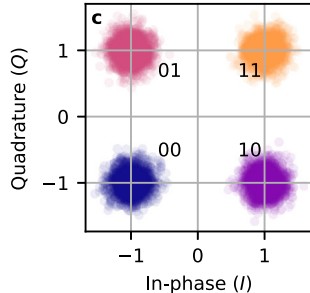

**Fig. 6 | Demonstration of a quadrature-amplitude modulation data transfer using the all-optical superhet. a** Power spectrum of the received signal before compensation ($S_{z_S}$) and after phase compensation ($S_{z_C}$). The phase compensation allows full recovery of the modulation features. Without the compensation, the features are blurred due to laser phase noise. Signal power is given with respect to the maximum compensated signal power. **b** Example transmission of 10 symbols at 122.07 kBaud rate. Solid curves (I and Q) correspond to value in each time bin, which is estimated from raw sample data (semi-transparent curves). Each symbol is estimated from 32 subsequent samples. **c** IQ diagram of received QAM4 symbols obtained from $8 \times 10^3$ symbols. Colours correspond to the original sent symbol, demonstrating a lack of errors in this transmission.

## Methods

### Theoretical model

To facilitate the comparison with the theoretical predictions, we prepare a numerical simulation based on the following considerations. The atomic state time evolution is described by the Gorini–Kossakowski–Sudarshan–Lindblad (GKSL) equation,

$$\partial_t \hat{\rho} = \frac{1}{i\hbar}[\hat{H}, \hat{\rho}] + \mathcal{L}[\hat{\rho}], \qquad (8)$$

where $\hat{H}$ is Hamiltonian and $\mathcal{L}[\cdot]$ is a superoperator responsible for sources of decoherence, such as spontaneous emission. We consider the steady state solution to the GKSL equation, $\partial_t \hat{\rho}(t) = 0$. Atomic numerical data, such as state lifetimes and transition dipole moments, is found via the Alkali Rydberg Calculator[49]. To get full agreement with the experiment, these operations have to be done for a range of velocity classes present in a room-temperature atomic medium, as well as for a range of laser field power, changing with the radial position in the Gaussian beams.

### Details of the optical-bias setup

The MW detection part of the optical-bias setup employed in the experiment is pictured in Fig. 2b. Four optical Gaussian beams are all focused with Gaussian waists $w_0 = 250$ μm inside a rubidium vapour cell. The rubidium is in natural abundance isotope proportion, though only $^{85}$Rb is addressed in this work. The optical length of the cell is 50 mm and it is kept in room temperature conditions, $T = 22.5\,°C$. The matched optimised Rabi frequencies of the fields (at the beam centres) are respectively $\Omega_P = 5.5 \cdot 2\pi$ MHz, $\Omega_C = \Omega_{C1} = 7.5 \cdot 2\pi$ MHz, $\Omega_D = 6.2 \cdot 2\pi$ MHz and $\Omega_{C2} = 9.5 \cdot 2\pi$ MHz for 780 nm, 480 nm, 776 nm and 1258 nm fields. All of the fields have matched circular polarisations. The lasers are stabilised via either cavity transfer locks or optical phase-locked loop to a common reference source (a narrowband fibre laser at 1560 nm, frequency-doubled to 780 nm and referenced to Rb cell with modulation-transfer lock). This results in collective spectral stability of around 62 kHz, as seen in Fig. 3a, b.

Near the vapour cell acting as a detector, we have placed a MW helical antenna acting as a source of a weak MW field (and additionally also as a source of MW LO in the superhet measurements facilitated for comparisons). The antenna emits circularly-polarised MW field collinearly with the optical fields. The collinear configuration ensures that the transduced signal contributions along the atomic medium interfere constructively, providing the best transduction efficiency. Other geometrical configurations are also possible, and in general, we have to consider the spatial averaging of the transduced signal that leads to an antenna pattern of the receiver (for a more detailed discussion, see

including calculated antenna pattern for our configuration, see Supplementary Section S.5).

Both the antenna and the vapour cell are placed inside a MW absorbing shield (made from LeaderTech EA-LF500 material), protecting from MW reflections and on-air noise. The probe laser beam counter-propagates to all the other fields, and its power is registered on an avalanche PD (Thorlabs APD430A). This detection is shot-noise limited with respect to the probe field power.

### Details of the laser phase detection setup

The optical beams 480 nm and 776 nm of respective powers ~ 100 mW and ~ 15 mW are focused and Rayleigh length-matched in a z-cut 30 mm MgO:PPLN crystal with 5.17 μm poling period, at ~ 85 °C, resulting in 1258 nm DFG signal at ~ 25 μW. We combine the DFG signal with ~1 mW of sampled 1258 nm laser field (which is also propagated through the crystal to reduce the differences in optical paths). Both of the 1258 nm fields are then fibre coupled. The beat-note between them is detected with a fast PD (25G SFP28 module for 1310 nm) and serves as an electronic MW phase reference signal.

The phase reference from DFG PD is downmixed with the LO to a frequency of $\Delta' = 4.6$ MHz convenient for Red Pitaya STEMlab 125-14 ADC/FPGA board used as a measurement tool. The signal is digitised and digitally demodulated, thus providing a complex signal $z_R(t)$. The uncompensated signal $\mathcal{I}_S$ is also IQ-demodulated at its central frequency $\Delta$, yielding complex time dependence $z_S(t)$. The compensated signal is then retrieved as:

$$z_C(t) = z_S(t)z_R^*(t), \qquad (9)$$

where with * we denote complex conjugation. Both the IQ mixers and the complex multiplication are implemented directly in the FPGA architecture. We calculate the processing latency (delay) by analysing the particular FPGA implementation, obtaining 72 ns for the ADC, and 96 ns for the IQ mixers and multiplication overall. Spectra of the uncompensated signal, reference phase signal, and compensated signal are presented in Fig. 3.

### Details of the MW setup and calibration

The MW frequency resonant with the $55^2D_{5/2} \rightarrow 54^2F_{7/2}$ transition is measured as $\omega_0 = 13912.4$ MHz via A-T splitting. The MW signal is then set at $\omega_0 + \Delta$, where $\Delta = 1.8$ MHz. This offset frequency is chosen large enough to avoid any electronic problems at low frequencies and fit within the detection bandwidth, resulting in a well-resolved sideband on the probe field (for the full consideration of the choice of frequencies, consult Supplementary Section S.1). The MW signal power is attenuated with a programmable MW attenuator (MiniCircuits RCDAT-

18G-63). All MW fields are calibrated using the A-T splitting method in the resolvable regime.

In the case of superhet detection, facilitated for comparison, the MW LO is added with a power splitter inserted in the antenna cable. The MW LO is resonant with the atoms and at $800\,\mu$V/cm ($1.6 \cdot 2\pi$ MHz Rabi frequency), which is measured experimentally as optimal in terms of SNR for the set detection parameters, and then combined with the MW signal at the antenna.

All the MW signals are generated with separate LMX2529 PLL frequency synthesizers synchronised to the same clock reference. The signal from the avalanche PD is measured with the STEMlab 125-14 and digitally demodulated, yielding a complex amplitude and phase signal.

### Carrier recovery in QAM transmission

Typically, during a quadrature-modulated transmission, the transmitter and receiver are not phase-stable with respect to each other. A technique of carrier phase recovery is therefore required. Here, we employ a technique called multiply-filter-divide, adopted for our situation. As the modulation has 4 symbols, we take the fourth power of the registered signal $z_C^4$, which effectively cancels out the modulation, allowing us to estimate the phase of the carrier as $\arg(z_C^4)/4$. We do this by averaging $z_C^4$ over 20 subsequent symbols. This allows us to compensate for the relative phase drift between transmitter and receiver, which is in part due to the transmitter and receiver using different clocks (quartz oscillators), but also due to interferometric stability (fluctuations of the $\phi_i$ phase) of the phase referencing chain. The requirement for this technique is that the baud rate is faster than the stability of the system, which is very strongly satisfied.

## Data availability

The data supporting the results presented in this paper are available via Harvard Dataverse[50].

## Code availability

The codes used for the numerical simulation and the analysis of experimental data are available from the corresponding author upon request.

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

## Acknowledgements

We thank K. Banaszek for the generous support. The "Quantum Optical Technologies" (MAB/2018/4, 2018–2023) project was carried out within the International Research Agendas programme of the Foundation for Polish Science, co-financed by the European Union under the European Regional Development Fund (2018–2021). The "Quantum Optical Technologies" (FENG.02.01-IP.05-0017/23, 2024–2029) project is currently carried out within the Measure 2.1 International Research Agendas programme of the Foundation for Polish Science, co-financed by the European Union under the European Funds for Smart Economy 2021–2027 (FENG). This research was funded in whole or in part by the National Science Centre, Poland, grant no. 2021/43/D/ST2/03114. S.B. is a recipient of Foundation for Polish Science START 2025 scholarship.

## Author contributions

S.B. built the optical, microwave, and electronic processing setups, assisted by other authors. W.W. developed the measurement system firmware and software, assisted by S.B., who programmed the software. S.B. took the measurements and analysed the data, assisted by other authors. S.B., M.M. and M.P. prepared the figures and the manuscript. W.W. developed the GKSL numerical simulation assisted by S.B., who facilitated comparison with experiments. M.P. conceived the central idea and led the project, assisted by M.M. and W.W.

## Competing interests

The authors declare no competing interests.
