## [Transparent Peer Review file · Nature Communications]

Optically-biased Rydberg microwave receiver enabled by hybrid nonlinear interferometry

Corresponding Author: Dr Michał Parniak

Version 0:

Reviewer comments:

Reviewer #1

(Remarks to the Author)

In this manuscript Borowka and coworkers demonstrate and describe an “optically biased” Rydberg receiver that obtains phase stable RF signal without needing to lock the optical phases. This work is nice and brings field closer to the important goal of simple all-optical receiving. I expect this manuscript can be of interest to a broad readership, though I have a few larger concerns/suggestions I would ask authors to address as well as a few smaller things further below:

-- One of the main reasons that phase-sensitive all-optical detection is desired is to receive modulated communications signals. This approach is an elegant workaround to an otherwise technically difficult problem (phase-locking several lasers) and I'm convinced that it does/should work. But beyond the spectral narrowing and the spectra in figure 5, I think the results would be much more convincing if there was a practical demonstration of the benefits of this approach such as detecting a modulated signal or some other evidence of longer-term phase stability.

-- While I do not find anything significantly wrong with it, I find the theoretical description of the approach (pages 2-3) clunky and hard to follow. I would suggest that the authors clarify/streamline this discussion a bit. As written, this section also glosses over some of the nuance inherent to Rydberg sensing that a broad readership may need to understand the process. For instance, the cosine term in equation 1 [$\cos(\Delta \cdot t)$] should also contain the phase difference of the LO and signal fields – this is the fundamental reason that the phase stability/correction is needed.

- I'm having a hard time following the purposes and implications of several of the frequencies described in the implementation section. What is the purpose of the Δ_{5D} detuning? If my interpretation is correct, it does not affect the beat modulation. I also wonder about the downconverted beat frequency of 4.6 MHz - why is it inadequate to mix to DC/baseband? I assume this also requires establishing and tracking the phase at this frequency?

- Since figure 1b is drawn with an IQ mixer, I am wondering if the authors tried using an analog RF circuit rather than the FPGA?

- In the spectra in figure 5, I'd be curious to learn a bit more about the peak structure of the optical bias/theory spectra. In particular, the peak separation appears to be comparable to the optical rabi frequencies, so I'm wondering how that relates?

- The language around sensitivity is in general challenging because a better sensitivity is a lower measurable field strength and vice versa. I would recommend changing the wording around “lower sensitivity” to avoid this ambiguity.

Reviewer #2

(Remarks to the Author)

Thank you for giving me the opportunity to review and study. I have carefully read the manuscript. The authors have achieved all-optical detection, which has very important research value. However, we think this manuscript is not suggested to be published in Nature communication without major revision.

Details are as follows:

1. In Figure 3c, why the reference signal is 4.6 MHz, while the reference signal in the conference PPT is about 6 MHz. The article doesn't explain in detail how to obtain the beat signal of 0 Hz.

2. The loop energy level structure shown in Figure 2 seems to be only for a specific discrete microwave frequency. Can this loop energy level structure be used for detuning microwave fields? If only a discrete frequency point microwave can be received, its application value is limited. Could the paper include additional experimental results related to this aspect?

3. The phase is processed in real-time by FPGA. Is there any relevant phase measurement result? The real-time performance of phase measurement will affect the instantaneous bandwidth of this method, which is another key indicator

apart from sensitivity. Are there any relevant experimental results?

4.NIST [1] has a similar experimental scheme. This manuscript only provides another specific implementation form. The most important thing is that its performance is similar to the existing methods. What are the innovative aspects of the method in this paper compared with it?

[1] S. Berweger, A. B. Artusio-Glimpse, A. P. Rotunno, N. Prajapati, J. D. Christesen, K. R. Moore, M. T. Simons, and C. L. Holloway, "Closed-loop quantum interferometry for phase-resolved rydberg-atom field sensing," *Physical Review Applied*, vol. 20, p. 054009, Nov. 2023.

Reviewer #3

(Remarks to the Author)

The paper by Barwoka et al. presents a method based on mixing to compensate for phase fluctuations in a closed loop Rydberg microwave receiver. The limitation is that the mixing requires a radio frequency source at the sensing frequency, which the authors point out and address. Data is presented to experimentally demonstrate the method and compare it to a superhet. The paper is interesting and should be published. The authors should go back through the paper and carefully edit it. In some places, such as the figure 3 caption, ex. 'chime'?, it is difficult to read. I have a few comments that should be addressed before recommending publication in Nature Communications.

1. The authors should mention the other Doppler free works that have been done recently in cesium. I don't think the paper and its point is restricted to rubidium. Some of those are also optical and quote better sensitivity.
2. Can the authors comment on the spatial aspects of the phase and how it affects the technique?
3. Can the authors add error bars to Figure 4 and/or quote the uncertainty in the figure. Also, can they explain what they mean by norming to the difference in the noise floor? Is this just the difference in the probe laser powers used or is there more to it?
4. Can the authors give error bars for the phase in Figure 5? I don't understand the color scheme - there is no color map that I saw.
5. Can the authors justify the claim that they are probe laser shot noise limited? A calculation would be nice.

Version 1:

Reviewer comments:

Reviewer #1

(Remarks to the Author)

I would like to thank the authors for their thorough reply. I am happy to see the included modulation data and the constellation diagram - I think this manuscript is ready for publication.

Reviewer #2

(Remarks to the Author)

The authors have addressed most of the issues raised. There are still a couple of issues that need to be explained.

Details are as follows:

1. The method in the manuscript also requires the use of a local oscillator microwave field. Therefore, the claim of all-optical measurements may still not be accurate enough. Please provide a detailed statement in the introduction.
2. Supplementary materials demonstrate the ability of this scheme to receive near-detuned QAM-modulated signals. We are also interested in the case of an incident far-detuned microwave field. Please discuss it in detail.
3. As shown in Figure 4, the manuscript only provides the sensitivity comparisons at specific energy levels (55D5/2-54F7/2). To determine the universality of this method, please provide a comparison of sensitivity and dynamic range for different Rydberg states.

Reviewer #3

(Remarks to the Author)

The authors have made all the suggested corrections. I recommend publication.

Version 2:

Reviewer comments:

Reviewer #2

(Remarks to the Author)

The author has clarified the issues. I think this manuscript is ready for publication.

Reviewer #1

In this manuscript Borowka and coworkers demonstrate and describe an “optically biased” Rydberg receiver that obtains phase stable RF signal without needing to lock the optical phases. This work is nice and brings field closer to the important goal of simple all-optical receiving. I expect this manuscript can be of interest to a broad readership, though I have a few larger concerns/suggestions I would ask authors to address as well as a few smaller things further below:

We thank the Reviewer for insightful suggestions and the positive assessment of our work. We hope that addressing the comments of this review will improve the manuscript. Please find the point-by-point responses below.

-- One of the main reasons that phase-sensitive all-optical detection is desired is to receive modulated communications signals. This approach is an elegant workaround to an otherwise technically difficult problem (phase-locking several lasers) and I'm convinced that it does/should work. But beyond the spectral narrowing and the spectra in figure 5, I think the results would be much more convincing if there was a practical demonstration of the benefits of this approach such as detecting a modulated signal or some other evidence of longer-term phase stability.

We thank the reviewer for acknowledging the elegance and impact of our result. We agree that while our experimental plots show the most essential performance of the system, they may not speak directly to readers interested in applications. This has prompted us to follow the Reviewer's suggestion and directly demonstrate a QAM transmission of a signal, which has now been added in the Results section. In particular, we are now able to show how QAM symbols are retrieved thanks to our phase correction method, which operates independently and in real-time.

As an additional example, we also show modulated spectra with and without the phase correction, elucidating the dramatic difference.

We also address the long-term stability issue. It is important to note that for a successful transmission, we require the phase to be stable over several symbols at least, but not an entire transmission. This is typical in classical transmitter/receiver systems, which typically do not share a common clock. To address this, both our system and classical systems employ carrier and/or clock recovery algorithms, which we now briefly discuss in the manuscript.

Finally, to explicitly evaluate the stability in the Supplementary Section S.3 we calculate Allan deviation, which reaches 2.1 Hz at 1.4 second averaging time. For the microwave frequency, this translates to less than 1 ppb. This is significantly better than required for a successful transmission with baud rates down to well below 1000 symbols per second. Transmissions at such low baud rates are, of course, relevant only with bad SNR, e.g. at a very long range.

-- While I do not find anything significantly wrong with it, I find the theoretical description of the approach (pages 2-3) clunky and hard to follow. I would suggest that the authors clarify/streamline this discussion a bit. As written, this section also glosses over some of the nuance inherent to Rydberg sensing that a broad readership may need to understand the

process. For instance, the cosine term in equation 1 [$\cos(\Delta \cdot t)$] should also contain the phase difference of the LO and signal fields – this is the fundamental reason that the phase stability/correction is needed.

We have thoroughly reworked the theoretical section for better readability and clarity. We paid special attention to the clarity of the presented derivations and tracking of the phase components.

- I'm having a hard time following the purposes and implications of several of the frequencies described in the implementation section. What is the purpose of the Δ_{5D} detuning? If my interpretation is correct, it does not affect the beat modulation. I also wonder about the downconverted beat frequency of 4.6 MHz - why is it inadequate to mix to DC/baseband? I assume this also requires establishing and tracking the phase at this frequency?

Concerning the Δ_{5D} detuning, this is right that this frequency does not affect the beat modulation. The purpose of this detuning is to slightly shift the centre of the receiving band to the chosen 1.8 MHz frequency. This action enhanced the received signal a little (by an order of 2-3 dB). For small detunings Δ , we discovered that these values follow the relation $\Delta_{5D} = -\Delta$ to achieve the best signal. This is, however, not strictly true for larger detunings. We now present in the Supplement Section S.4 the bandwidth measurement for $\Delta_{5D} = 8$ MHz, where the centre of the band lands at 9.5 MHz.

We have now provided the Supplementary Section S.1 to expand on the motivation behind the choice of the other frequencies. Furthermore, Fig. 1 is modified to show how various frequencies interact.

Setting the registered frequencies to DC (baseband) is impractical due to the noise at low frequencies (probe field intensity fluctuation due to transduction of frequency noise over narrow EIT features and electronic noise). This is now shown in the expanded figures in the Supplement — one can see significantly increased noise, especially below 100 kHz. This would also prevent us from obtaining the full complex signal, and we would need to use an analogue IQ mixer.

The specific choice of the reference frequency of 4.6 MHz is due to technical reasons, i.e. to optimize between the best rejection of higher order modulation and not interfering with the signal frequency.

In our digital IQ mixing stage, both the signal and the reference frequencies are demodulated to low frequencies for full phase processing. As mentioned, we now present the expanded scheme explaining that in the frequency domain in Fig. 1.

- Since figure 1b is drawn with an IQ mixer, I am wondering if the authors tried using an analog RF circuit rather than the FPGA?

We thank the reviewer for this insightful idea.

First of all, we would like to stress that now we have decided to put a much more detailed depiction of the setup in the main manuscript. We hope that it will help with understanding the operational ideas behind the presented circuit.

Indeed, we agree that the realisation could be fully analog, with ADC only at the very end. However, when implementing the mixing operations in the analog circuit, we need to take into account image rejection, proper signal levels, and filtering. The FPGA-based approach allows for versatility without the loss of any sensitivity. We are able to easily implement high-order spectral filtering in real-time if necessary. Our system allows for changing filters and mixers without synthesising a new FPGA circuit. Furthermore, the delay due to digital processing is small, of the order of 100 ns.

Below, we present an example all-analog realisation of the same idea. This proposal involves upmixing the atomic signal and then mixing the result with the amplified reference signal. Finally, the baseband heterodyne signal (at the MHz-level offset frequency) is converted to digital data, in the same way as the normal superhet.

Both in the superhet and this example, one could also use a fully-analog IQ mixer at the end of the setup, and only then proceed with an ADC. We believe such a solution would have worse performance, as the ADC would be impacted by low-frequency noise.

Overall, we strongly believe that our solution, which heavily relies on DSP, brings many advantages over the more traditional analogue solutions. Our approach is more in line with modern electronics and SDR-type solutions (software-defined radio). We outline those advantages in the main text, but we chose not to discuss the analog alternatives further, as we feel that a comparison of many alternatives is too technical beyond the scope of this paper.

- In the spectra in figure 5, I'd be curious to learn a bit more about the peak structure of the optical bias/theory spectra. In particular, the peak separation appears to be comparable to the optical Rabi frequencies, so I'm wondering how that relates?

In the simplest case of a three-level atom, the shape of the EIT peak is indeed straightforwardly defined by the optical Rabi frequency and state lifetime (and Doppler effect) — but this is not always the case in more complex multilevel schemes. Furthermore, we agree that the EIT peak separation in the optical-bias method is interesting, especially as it bears similarities to the simpler A-T splitting observed with probe and coupling fields (where peak separation is proportional to the Rabi frequency of the microwave LO field). However, the similarity stays true only in the optimal detection point — we confirmed both in experiment and in simulations that a

different choice of Rabi frequencies for optical fields (in particular the 776 nm field) leads to very different shapes of EIT, where more resonances can be seen. It could be exciting to study these effects in the dressed-state formalism and maybe arrive at the conclusion of whether the similarities in the EIT shape seen via probe absorption are coincidental, or if they in fact result from mimicking the similar effect by preparing the atoms in a similar state. While it is indeed true that the peak separation in the optical-bias method is of the order of optical Rabi frequencies, the complexity of the system, especially the role of different velocity classes, prevents us from drawing a simple conclusion. Therefore, we consider further analysis of this complexity beyond the scope of the current work.

- The language around sensitivity is in general challenging because a better sensitivity is a lower measurable field strength and vice versa. I would recommend changing the wording around "lower sensitivity" to avoid this ambiguity.

We thank the Reviewer for this suggestion. We have now reviewed the manuscript and changed the wording so that the adjectives referring to sensitivity unambiguously address the performance rather than its level.

Reviewer #2

Thank you for giving me the opportunity to review and study. I have carefully read the manuscript. The authors have achieved all-optical detection, which has very important research value. However, we think this manuscript is not suggested to be published in Nature communication without major revision.

We thank the Reviewer for acknowledging the importance of our experimental work and for spotting the issues that required revision. Please find the responses to the comments below.

Details are as follows:

1. In Figure 3c, why the reference signal is 4.6 MHz, while the reference signal in the conference PPT is about 6 MHz. The article doesn't explain in detail how to obtain the beat signal of 0 Hz.

We thank the Reviewer for attending our talk at the mentioned conference. Indeed, in some presentations we have shown preliminary results, where the reference was centred at 6 MHz. However, for final measurements that prioritised sensitivity and fair comparison with the superhet method, we have moved the reference to 4.6 MHz for purely technical reasons (to optimise between the best rejection of higher order modulation and no interfering with the signal frequency). The work presented in the manuscript consistently uses the 4.6 MHz frequency for the reference.

We now provide greater detail on describing the choice of frequencies (please see the Supplementary Section S.1) and on how to obtain the demodulated signal at 0 Hz.

2. The loop energy level structure shown in Figure 2 seems to be only for a specific discrete microwave frequency. Can this loop energy level structure be used for detuning microwave fields? If only a discrete frequency point microwave can be received, its application value is limited. Could the paper include additional experimental results related to this aspect?

We agree that the initial version of the manuscript did not sufficiently demonstrate the ability to receive detuned fields or the whole bandwidth. To demonstrate direct evidence of receiving a wideband signal, we performed additional measurements and we now provide data for receiving and full demodulation of a QAM-modulated signal in the Results section. Please also see Supplementary Section S.4 for the considerations regarding the receiving bandwidth for detuned fields, based on additional measurements.

3. The phase is processed in real-time by FPGA. Is there any relevant phase measurement result? The real-time performance of phase measurement will affect the instantaneous bandwidth of this method, which is another key indicator apart from sensitivity. Are there any relevant experimental results?

We thank the Reviewer for the comment, as it has prompted us to better explain the method, its versatility, and present additional data. We have now reworked and expanded the operational description on how the signal is processed in the FPGA, including the retrieval of the phase in the Results section.

In the initial submission, the phase retrieval via real-time processing was already present in Fig. 5. Now we also present additional results concerning the full phase retrieval with QAM demodulation — in the Results section, in particular Fig. 6.

Furthermore, we have now performed the analysis of long-term measurement and calculation of Allan variance, in the Supplementary Section S.3, which points to the limit on the achieved phase stability.

In the case of employed real-time processing, the set bandwidths are controlled digitally. The latency in this case does not affect bandwidth, as there is no feedback in the real-time processing, only feedforward. Thus, the bandwidth in our measurements in most cases is limited only by the digitally set boundaries. To investigate the bandwidth limitations imposed by the atomic properties, we now present an additional bandwidth measurement (with a different configuration of digital settings to not be limited by the set processing bandwidth) for the optical-bias method, in the Supplementary Section S.4.

4. NIST [1] has a similar experimental scheme. This manuscript only provides another specific implementation form. The most important thing is that its performance is similar to the existing methods. What are the innovative aspects of the method in this paper compared with it?

[1] S. Berweger, A. B. Artusio-Glimpse, A. P. Rotunno, N. Prajapati, J. D. Christesen, K. R. Moore, M. T. Simons, and C. L. Holloway, "Closed-loop quantum interferometry for phase-resolved rydberg-atom field sensing," *Physical Review Applied*, vol. 20, p. 054009, Nov. 2023.

We thank the Reviewer for bringing up this important work. First of all, we would like to underline that the work from NIST has already been acknowledged and referenced as Ref. [42] (Ref. [38] in our original submission). In that work, a loop utilising two microwave transitions and two optical transitions was realised, where, in a practical scenario, only one of the microwaves is used as a signal, while the other acts as an auxiliary field. In contrast, our work does not use any auxiliary microwave fields in the detection subsystem, and thus, the detection is all-optical. This difference is not merely technical, as it allows one to completely avoid the usage of any auxiliary microwave antennas, emitters, waveguides, or electronics. As seen in our work, the all-optical operation brings about challenges of its own, with laser phase noise being the primary one, which we solve in our work.

The authors in the Ref. [42] compare their method to the superheterodyne detection method (where signal and field are addressing the same transition), and their method has sensitivity 10 dBV lower than superhet, while ours is only 3 dBV lower. In that regard, in practical demonstration, our method can be considered more sensitive, in addition to its all-optical operation. Achieving sensitivity comparable to superheterodyne detection, while utilising only optical fields, is first demonstrated in the present manuscript and is an improvement of several orders of magnitude compared to the previous all-optical methods (based on A-T splitting), while the full phase readout is a completely new feature. The authors in the Ref. [42] also suggest several other all-optical methods to be realised (a little more complex than ours); however, the problem of laser phase stability, which we overcame, is not addressed there. Clearly, there has also been no previous experimental demonstration.

Reviewer #3

The paper by Barwoka et al. presents a method based on mixing to compensate for phase fluctuations in a closed loop Rydberg microwave receiver. The limitation is that the mixing requires a radio frequency source at the sensing frequency, which the authors point out and address. Data is presented to experimentally demonstrate the method and compare it to a superhet. The paper is interesting and should be published. The authors should go back through the paper and carefully edit it. In some places, such as the figure 3 caption, ex. 'chime'?, it is difficult to read. I have a few comments that should be addressed before recommending publication in Nature Communications.

We thank the Reviewer for showing interest in our work and underlining the main advantages of the new method. We have followed the suggestion and thoroughly revised the text of the paper to increase its readability. Please find the responses to the other comments below.

1. The authors should mention the other Doppler free works that have been done recently in cesium. I don't think the paper and its point is restricted to rubidium. Some of those are also optical and quote better sensitivity.

We thank the Reviewer for this comment, as we were aware of this work, but we must have missed the opportunity to refer to it. We now provide the reference (Ref. [47]) and expand on the topic in the discussion. The Doppler-free Autler-Townes splitting studies are indeed very valuable and offer excellent sensitivity. The A-T type detection however, is not directly phase sensitive, and thus is not well-suited for data transmission. For the phase sensitivity, it would again require a microwave LO. We trust in future opportunities to combine the Doppler-free schemes with our all-optical mixer scheme. We have also expanded the references to include more of the applicable results in caesium.

2. Can the authors comment on the spatial aspects of the phase and how it affects the technique?

We thank the Reviewer for this insightful comment. The phase of the probe beam modulation arising from the presence of the signal microwave field can indeed exhibit spatial dependence. Due to spatial averaging — or more precisely, phase matching — this spatial variation affects the efficiency of the optical readout and leads to a reception (antenna-like) pattern of the sensor. This applies to both the superheterodyne and the optical-bias methods, and becomes important when the spatial period $1/\delta k_z$ (where δk_z is the phase mismatch along the optical propagation axis) is comparable to the effective interaction region length L .

In the superheterodyne method, since the phases of the probe and coupling beams cancel out, the spatial phase is determined by the difference in wavevectors of the signal and LO microwave fields $\delta k_z = k_z^\varepsilon - k_z^{LO}$, which is zero in the typical case of copropagating fields.

In the optical-bias method, the spatial phase includes contributions from the biasing optical fields and the signal microwave field. For large optical beams ($z_R \gg L$), the mismatch can be approximated as $\delta k_z = k_z^{480} + k_z^\varepsilon - k_z^{1258} - k_z^{776}$. For small beam widths ($z_R \sim L$), the situation becomes more complex, as the phase mismatch also depends on the transverse coordinates.

In our implementation, both methods are operated in collinear configurations, for which $\delta k_z \approx 0$. A comprehensive analysis of the spatial phase for varying signal incidence angles and the resulting reception patterns under various experimental configurations would certainly be interesting, but we believe this lies beyond the scope of the present work.

We trust that this is the most essential spatial aspect of the reception process. A more detailed discussion, including the calculated reception pattern for our experimental conditions, is now included in the supplementary material (Supplementary Section S.5).

3. Can the authors add error bars to Figure 4 and/or quote the uncertainty in the figure. Also, can they explain what they mean by norming to the difference in the noise floor? Is this just the difference in the probe laser powers used or is there more to it?

In the original version of Fig. 4, the uncertainties were not included, as in most cases they were smaller than the size of the data points. In the new version of Fig. 4, we now include the uncertainties as a separate subfigure displaying the relative standard error of the presented data. The normalisation to the respective noise floors is only a technicality due to different effective gains employed in the digital processing in the superhet and optical-bias methods, i.e. digital signals of different levels are registered in the FPGA. We now include this information in the manuscript. In both cases (superhet and optical-bias), the same probe laser power was used, as stated in the manuscript.

4. Can the authors give error bars for the phase in Figure 5? I don't understand the color scheme - there is no color map that I saw.

We agree that the original color scheme and shading of the plots were not clear. We therefore removed the gradient shading from the upper row presenting the probe transmission spectra. We also changed the color scheme for the phase plots in the lower row. The phase uncertainty is now marked as shaded regions representing the standard error of the displayed data.

Moreover, the plot labels have been updated to match the new version of the description in the manuscript.

5. Can the authors justify the claim that they are probe laser shot noise limited? A calculation would be nice.

We agree that the claim of the shot noise limit was previously not justified. We now provide the measurement of noise scaling (i.e. linear scaling of noise power with laser power, being a hallmark of shot noise) directly evidencing the dominance of the shot noise component. Please see the Supplementary Section S.2. We estimated the overall noise in the detection to contain 83% of the shot noise component. To make it clearer, we have also updated Fig. 4 to indicate the registered levels properly.

Additionally, we refer to the previous work that performed the calculation for shot noise in the superhet method, Ref. [44]. While we agree with this method, we consider the estimation of the shot noise via power scaling device-independent, as it does not need to rely on the declared detector efficiencies.

We also apologize, as the previous results were incorrectly normalized to total noise rather than just shot noise, so it was wrong by 17%. Now the reader can clearly distinguish the total noise, shot noise, and other (electronic) noise. The deposited data have also been corrected and supplemented.

We also now comment on the reason for our shot-noise limited sensitivity: both in the superhet and the optical-bias schemes, we choose the detunings Δ and Δ' such that they lie in the spectral region (a few MHz) where shot noise is dominant. This way we avoid low-frequency $1/f$ technical noise: e.g. excess photodiode electronic noise as

Response to reviewers' comments

NCOMMS-25-23694

Optically-biased Rydberg microwave receiver enabled by hybrid nonlinear interferometry

well as laser phase noise (dominant at low frequencies, as the laser linewidth is of the order of 20 kHz) transduced to amplitude noise via EIT.

Reviewer #1

I would like to thank the authors for their thorough reply. I am happy to see the included modulation data and the constellation diagram - I think this manuscript is ready for publication.

We thank the Reviewer for the positive assessment and recommendation for publication.

Reviewer #2

The authors have addressed most of the issues raised. There are still a couple of issues that need to be explained.

We thank the Reviewer for the acknowledgement of our addressing of the issues. Please find the responses to the new issues below.

Details are as follows:

1. The method in the manuscript also requires the use of a local oscillator microwave field. Therefore, the claim of all-optical measurements may still not be accurate enough. Please provide a detailed statement in the introduction.

We agree that the microwave local oscillator is required in practice if our method of laser phase noise compensation is employed. However, this requirement is strictly at the electronic processing stage (RF cables and instruments), so that the detecting stage (free space and atomic vapours) remains truly all-optical. Furthermore, if extremely stable lasers are used in the optical-bias detection presented in the manuscript, the need for the local oscillator can be neglected.

Nevertheless, following the suggestion, we have now better clarified the all-optical claim in the Introduction.

2. Supplementary materials demonstrate the ability of this scheme to receive near-detuned QAM-modulated signals. We are also interested in the case of an incident far-detuned microwave field. Please discuss it in detail.

While we agree that the case of far-detuned fields is of particular interest in terms of semicontinuously tunable all-optical sensing, we believe that full consideration is beyond the scope of this work. We present the capability of the new sensing method to detect QAM-modulated signals, which has also been previously demonstrated for the superhet method a few times. As both of the methods rely on wavemixing processes, in general, the methods for the detection of far-detuned fields can be translated from the superhet architecture. The main difference here is that there are more fields to be tuned to follow the signal field, similarly to tuning the LO in the superhet method. For the far-detuned MW fields, the main field to be detuned in the optical-bias detection would be the C2 coupling field, which would parallel the

detuning of the LO in the superhet method. This is certainly an interesting prospect, which could be explored further in future works. As shown, for example, in Ref. 10, the expected sensitivity indeed is worse for far-detuned signals.

The QAM-reception itself remains the same irrespective of whether the Rydberg receiver itself detects resonant (as in the main text), near-detuned (as in Suppl. Section 4), or far-detuned signals.

We have added this consideration in the Discussion section.

3. As shown in Figure 4, the manuscript only provides the sensitivity comparisons at specific energy levels ($55D5/2-54F7/2$). To determine the universality of this method, please provide a comparison of sensitivity and dynamic range for different Rydberg states.

Here, a similar argument follows, as the one to the second issue raised. As both the optical-bias method and the superhet method rely on a similar process of wavemixing (the similarities between both methods are further supported by the direct comparison performed in the same setup, where comparable sensitivities and dynamic ranges are obtained), it can be expected that the shift to different Rydberg transitions will follow the same relations. Numerous transitions have already been studied for the superhet method, and in particular, there are works focused on the differences between the transitions, such as in Ref. [11]. The main finding there is, quite predictably, that the transition dipole moment affects both the sensitivity and dynamic range due to the effect of scaling, while additionally, the choice of orbital momentum of Rydberg states also affects linearity for stronger fields, which may shift the saturation point in the superhet-type detection and thus affect the dynamic range. For higher- n states, Rydberg interactions may also influence the figures of merit. Due to the multitude of effects, at this stage, it is not possible to make accurate numerical predictions.

We have added this consideration with reference in the Discussion section.

Reviewer #3

The authors have made all the suggested corrections. I recommend publication.

We thank the Reviewer for the recommendation to publish.